# The heat shock protein 90 inhibitor RGRN-305 attenuates SARS-CoV-2 spike protein-induced inflammation *in vitro* but lacks effectiveness as COVID-19 treatment in mice

Hakim Ben Abdallah[1]*, Giorgia Marino[2], Manja Idorn[2], Line S. Reinert[2], Anne Bregnhøj[1], Søren Riis Paludan[2], Claus Johansen[1]

1 Department of Dermatology, Aarhus University Hospital, Aarhus N, Denmark, 2 Department of Biomedicine, Aarhus University, Aarhus C, Denmark

* hba@clin.au.dk

## Abstract

The inhibition of heat shock protein 90 (HSP90), a molecular chaperone, has been proposed to be a potential novel treatment strategy for Coronavirus disease 2019 (COVID-19). In contrast to other studies, our data demonstrated that RGRN-305, a HSP90 inhibitor, exacerbated the cytopathic effect and did not reduce the viral shedding in VeroE6-hTMPRSS2 cells infected with severe acute respiratory syndrome coronavirus 2 (SARS-CoV-2). Likewise in a murine model of SARS-CoV-2, transgenic mice treated orally with RGRN-305 exhibited reduced survival by the end of the experiment (day 12) as 14% (1/7) survived compared to 63% (5/8) of those treated with drug-vehicle. Animal weight was not reduced by the RGRN-305 treatment. Interestingly, we demonstrated that inhibition of HSP90 by RGRN-305 significantly dampened the inflammatory response induced by SARS-CoV-2 spike protein in human macrophage-like cells (U937) and human lung epithelial cells (A549). Measured by quantitative real-time PCR, the mRNA expression of the proinflammatory cytokines *TNF*, *IL1B* and *IL6* were significantly reduced. Together, these data suggest that HSP90 inhibition by RGRN-305 exacerbates the SARS-CoV-2 infection in *vitro* and reduces the survival of mice infected with SARS-CoV-2, but exhibits strong anti-inflammatory properties. This data shows that while RGRN-305 may be helpful in a 'cytokine storm', it has no beneficial impact on viral replication or survival in animals as a monotherapy. Further animal studies with HSP90 inhibitors in combination with an anti-viral drug may provide additional insights into its utility in viral infections and whether HSP90 inhibition may continue to be a potential treatment strategy for COVID-19 disease.

## Introduction

Coronavirus disease 2019 (COVID-19) is a viral respiratory contagious disease caused by the severe acute respiratory syndrome coronavirus 2 (SARS-CoV-2) [1]. The disease emerged as a worldwide pandemic causing a sudden global health crisis [2]. The clinical presentation of

**Data Availability Statement:** All relevant data are within the manuscript and its Supporting information files.

**Funding:** This study was supported by a research grant to HBA received from Aage Bangs Foundation. The funder had no role in the study design, data collection and analysis, decision to publish, or preparation of the manuscript.

**Competing interests:** I have read the journal's policy and the authors of this manuscript have the following competing interests: H.B.A has received research grants from Aage Bangs Fund and participated in clinical trials sponsored by Galderma, Regranion and UCB. C.J. has served as a paid speaker for LEO Pharma, Eli Lilly and L'Oréal This does not alter our adherence to PLOS ONE policies on sharing data and materials.

SARS-CoV-2 infection ranges from asymptomatic and mild upper respiratory symptoms to pneumonia and multiorgan failure, which can be fatal [3]. The pathophysiology of SARS-CoV-2 involves cell injury and death to virus-infected cells as a consequence of the viral replicative cycle. In some individuals, an excessive immune response may be triggered, involving a release of proinflammatory cytokines such as TNF, IL-1β and IL-6 that may inflict widespread lung and multiorgan hyperinflammation and damage [4,5]. The prophylactic vaccines for COVID-19 have been shown to be safe and effective against severe disease, hospitalization and death [6]. Nonetheless, vaccine hesitancy, varying vaccine efficacy and durability, breakthrough infections and the emergence of novel variants underline the need for therapeutic interventions that provide high protection against severe COVID-19 disease. Heat shock protein 90 (HSP90), a molecular chaperone that maintains cellular homeostasis by folding and assembling endogenous client proteins such as proinflammatory cytokines, has been shown to facilitate viral replication by folding viral proteins for multiple viruses (e.g. SARS-CoV-2, influenza A and herpes simplex virus-1) [7–9]. Several *in vitro* studies have shown that HSP90 inhibition may reduce the viral replication of SARS-CoV-2 as well as the excessive proinflammatory cytokine expression (also referred to as the "cytokine storm") [10–14]. Thus, recent studies from the literature suggest that HSP90 inhibition may be a potential novel therapeutic strategy for COVID-19 treatment.

In this study, we examined RGRN-305, a novel HSP90 inhibitor, as monotherapy in SARS-CoV-2-infected mice. Our hypothesis was that inhibition of HSP90 would impact both SARS-CoV-2 replication and proinflammatory cytokine expression, and thereby prolong survival in a murine model of SARS-CoV-2.

## Materials and methods

### HSP90 inhibitor RGRN-305

The HSP90 inhibitor RGRN-305 (molecular weight = 442.58 g/mol), formerly named Debio0932 and CUDC-305, and 17-allylamino-17-demethoxygeldanamycin (17-AAG) were purchased from MedChemExpress (HY-13469, HY-10211; Monmouth Junction, New Jersey, USA). RGRN-305 is a pan-inhibitor targeting HSP90α/β (IC50 ~ 0.1 μM), Grp94 (IC50 ~ 0.2 μM) and TRAP1 (IC50 ~ 1.5 μM).

### Experiments with SARS-CoV-2 virus *in vitro*

**SARS-CoV-2 propagation.** The Wuhan-Like, early European B.1 SARS-CoV2 isolate (provided by Professor Georg Kochs, University of Freiburg, FR-4286, GISAID accession no. EPI_ISL_852748) and alpha/B.1.1.7 variant SARS-CoV-2 (provided by Professor Arvind Patel, University of Glasgow, NCBI GenBank accession no. MZ314997) were propagated in VeroE6 cells expressing human TMPRSS2 (provided in early April 2020 by Professor Stefan Pöhlmann, University of Göttingen). The cell line was first described by Hoffmann *et al.* [15]. The VeroE6-hTMPRSS2 cells were infected with a multiplicity of infection (MOI) of 0.05 while maintained in a medium of Dulbecco's Modified Eagle Medium (Gibco, ThermoFisher Scientific, Massachusetts, USA) + 2% fetal calf serum (Sigma-Aldrich, Missouri, USA) + 1% Penicillin/Streptomycin (Gibco) + L-glutamine (Sigma-Aldrich). Following 72 hours post-infection, the supernatant with new virus progeny was harvested and concentrated on 100 kDa Amicon ultrafiltration columns (Merck, New Jersey, USA) by centrifugation at 4000 x $g$ for 30 minutes. Subsequently, the virus titer was quantified by Tissue Culture Infectious Dose (TCID$_{50}$) assay and the Reed-Muench method.

**RGRN-305 ED/LD50% analysis.** VeroE6 hTMPRSS2 cells were seeded 2 x 10$^4$ cells in Dulbecco's Modified Eagle Medium (Gibco) + 2% fetal calf serum (Sigma-Aldrich) + 1%

penicillin/streptomycin (Gibco) + L-glutamine (Sigma-Aldrich) to per well in a flat-bottomed 96-well plate. Twenty-four hours later, VeroE6-hTMPRSS2 cells were treated with RGRN-305 at concentrations of 0 μM, 0.1 μM, 1.0 μM and 10 μM (octuplicates) 1 hour prior to infection with B.1. (FR4286) isolate at MOI of 0.05 and 0.5 (triplicates). A plate of RGRN-305 treated cells were left uninfected to assess the toxicity (lethal dose 50) of RGRN-305 on VeroE6-hTMPRSS2 cells. 72 hours post infection (73 hours post treatment), in a 37˚C humidified 5% $CO_2$ incubator, the wells were fixed with 5% formalin (Sigma-Aldrich) and stained with crystal violet solution (Sigma-Aldrich). The wells were scored for the presence of cytopathic effect on the cells, and a % of the cytopathic effect was calculated. Images were taken using a Leica DMi1 microscope at 4x objective with a Leica MC170 HD camera.

**SARS-CoV-2 TCDI50% assay.** For assessing inhibition of release of new viral progeny from infected cells, a TCID50% as described above was performed. In short, $1x10^5$ VeroE6-hTMPRSS2 cells were seeded in 12-well plates containing 0.5 mL Dulbecco's Modified Eagle Medium + 2% fetal calf serum + 1% penicillin/streptomycin + L-glutamine. 24 hours after seeding, the cells were treated in biological triplicates with 0 μM, 0.01 μM, 0.1 μM or 1.0 μM of RGRN-305 or 17-AAG 1 hour prior to infection with B.1. (FR4286) isolate at 0.05 MOI. The virus was left to adsorb for 1 hour, after which the virus + RGRN-305/17-AAG/vehicle containing medium was removed and replaced with fresh RGRN-305/17-AAG/vehicle containing medium. The infection was allowed to run for 48 hours in a 37˚C humidified 5% $CO_2$ incubator, after which the medium containing released viral progeny was harvested.

Virus containing supernatants were titrated in 10-fold dilution series onto VeroE6-hTMPRSS2 cells– $2x10^4$ cells per well in a 96-well plate—and incubated for another 72 hours in a 37˚C humidified 5% $CO_2$ incubator. One full plate was used per sample. Each dilution ($10^{-1}$–$10^{-15}$) was represented in octuplicates. Infection and cytopathic effect were scored as described above, and TCDI50% virus titer was calculated by the Reed-Muench method.

## SARS-CoV-2 mouse model

Transgenic K18-hACE C57BL/6J mice (strain: 2B6.Cg-Tg(K18-ACE2)2Prlmn/J) expressing human ACE2 were obtained from The Jackson Laboratory (Stock number: 034860; Maine, USA). The mice were fed a standard chow diet and housed in a pathogen-free facility. Mice of the same age and both sexes were randomized to receive RGRN-305 (40 mg/kg, per os, [p.o.]) or vehicle (5% Kleptose HPB® Oral grade Roquette Pharma) once daily. The first dosage of RGRN-305 or vehicle was administered right after the intranasal inoculation with $1.5 \times 10^3$ plaque-forming units (PFU) of SARS-CoV-2. The virus inoculations were performed under anesthesia with an intraperitoneal injection of ketamine (100 mg/kg body weight) and xylazine (10 mg/kg body weight). The bodyweight of the mice was measured every day at the same time of the day until the end of the experiment (day 12 post-infection), death or reaching a humane endpoint (i.e., >20% bodyweight reduction, cessation of movement or respiratory distress). The mice were euthanized by cervical dislocation.

All animal experimental procedures were approved in advance by the Animal Ethics Committee at the Danish Veterinary and Food Administration (Stationsparken 31–33, 2600 Glostrup, Denmark). The study was carried out in accordance with the Danish Animal Welfare Act for the Care and Use of Animals for Scientific Purposes. All procedures followed the recommendations of the Animal Facility at the Aarhus University, and all efforts were made to minimize suffering.

## Experiments with SARS-CoV-2 spike protein *in vitro*

**A549.** The human lung epithelial cell line A549 was purchased from Merck (Darmstadt, Germany, Catalog#86012804) and seeded at a density of $1 \times 10^5$ cells/mL per well in 6 well-plates. The cells were cultured in Ham's F-12 Nutrient Mix (Gibco) supplemented with 10% heat-inactivated fetal bovine serum (Gibco) + 2 mM L-glutamine (Gibco) + 1% penicillin/streptomycin (Gibco) and incubated at 37°C and 5% $CO_2$ in a humidified incubator until 60–70% confluency. Subsequently, the medium was changed to a serum-reduced medium with 1% heat-inactivated fetal bovine serum for 24 hours before experiments were initiated.

**U937.** The pro-monocytic cell line U937 (kindly provided by Professor Martin Tolstrup, Aarhus University) was seeded at a density of $1.5 \times 10^5$ cells /mL per well in 6 well-plates and cultured in RPMI-1640 (Gibco) supplemented with 10% heat-inactivated fetal bovine serum (Gibco) + 1% penicillin/streptomycin (Gibco) + 0.05% gentamicin (Gibco) for 48 hours. Subsequently, the U937 cells were differentiated into macrophage-like cells by culturing in a medium containing 100 nM of 12-O-tetradecanoylphorbol-13-acetate (P8139; Sigma-Aldrich) for 48 hours. The U937 macrophage-like cells were washed twice with a pre-warmed medium and maintained in a medium without 12-O-tetradecanoylphorbol-13-acetate for the next 24 hours before experiments were initiated.

**Cell experiments.** The A549 and U937 macrophage-like cells were preincubated with RGRN-305 for 8 hours before stimulation with wild-type SARS-CoV-2 spike (S) protein (ThermoFisher; Catalog#: RP87668). The A549 cells were stimulated with 1000 ng/mL SARS-CoV-2 S protein for 24 and 48 hours, whereas the U937 macrophage-like cells were stimulated with 500 ng/mL and 5000 ng/mL SARS-CoV-2 S protein for 8 and 24 hours. In preliminary experiments (n = 2), four concentrations of the SARS-CoV-2 S protein (100, 500, 1000, and 5000 ng/mL) were tested. The optimal concentrations were selected based on their ability to induce inflammatory gene expression and minimize cell toxicity. The RNA was isolated, and the expression of inflammatory cytokines was measured by reverse-transcription quantitative PCR (RT-qPCR).

**RNA isolation.** The cells were washed twice with ice-cold phosphate-buffered saline (Gibco) and SV RNA lysis buffer (Promega, Madison, Wisconsin, USA)) was added. Total RNA was isolated using SV 96 Total RNA Isolation system (Promega) by following the manufacturer's instructions.

**Reverse transcription-quantitative PCR.** Total RNA was reverse transcribed to cDNA using random hexamers and TaqMan™ Reverse Transcription Reagents (Thermofisher) according to the manufacturer's instructions. Real-time PCR was performed with StepOnePlus Real-Time PCR system (Applied Biosystems™) using TaqMan™ Universal PCR Master Mix (ThermoFisher) and TaqMan™ primers and probes for *TNF* (Hs00174128_m1), *IL1B* (Hs01555410_m1), *IL6* (Hs00174131_m1) and *GAPDH* (Hs02786624_g1). The expression of *TNF*, *IL1B* and *IL6* was normalized to the reference gene *GAPDH*, and the normalized expression values were shown as fold changes relative to the levels in unstimulated and untreated samples (vehicle).

## Enzyme-linked immunosorbent assay (ELISA)

The secreted protein levels of TNF, IL-1β and IL-6 in cell culture supernatant were measured using commercial ELISA development kits (#DY201, #DY206, #DY210; Bio-Techne, Minnesota, USA) and a microplate spectrophotometer (Multiskan GO, ThermoFisher) following the manufacturer's protocol. All measurements were performed in duplicates.

## Statistical analyses

The pairwise difference of means in mRNA expression was analyzed with two-tailed paired T-test if data were normally distributed, otherwise Wilcoxon signed-rank test was used. The mean relative weight changes of mice were analyzed by 2-way ANOVA followed by Sidak's multiple comparisons. The survival curves were compared by a log-rank test (Mantel-Haenszel method). All statistical analyses were performed in GraphPad Prism V9.0 (GraphPad Software, California, USA). P-values < 0.05 were considered statistically significant.

# Results

## Cytopathic effect of SARS-CoV-2 in VeroE6-hTMPRSS2 cells may be exacerbated by RGRN-305

To examine the antiviral effects of HSP90 inhibition, VeroE6-hTMPRSS2 were treated with increasing concentrations of RGRN-305 (0.1 μM, 1 μM and 10 μM) or ionized water (drug-vehicle) and infected with SARS-CoV-2 virus at MOI 0.05 or 0.5 (Fig 1). RGRN-305 was toxic to the uninfected cells at 10 μM, whereas 0.1 μM and 1 μM were tolerated. The toxicity of RGRN-305 was increased by SARS-CoV-2 infected cells (both at MOI 0.5 and 0.05). In the 1 μM wells, no toxicity was detected with the drug alone, but in combination with virus at MOI 0.5, cells die in a visually similar manner to the cells treated with 10 μM. The 0.1 μM had no toxic effect on infected cells. However, none of the tested concentrations of RGRN-305 resulted in inhibition of virus induced cytopathic effects. Thus, these data indicate that HSP90 inhibition may exacerbate the cytopathic effect of SARS-CoV-2 in VeroE6-hTMPRSS2 cells.

## HSP90 inhibition by RGRN-305 or 17-AAG increased the SARS-CoV-2 replication in VeroE6-hTMPRSS2 cells

The effect of HSP90 inhibition by RGRN-305 on the viral shedding of infected cells was explored with 17-AAG used as a comparative reference. The VeroE6-hTMPRSS2 cells were

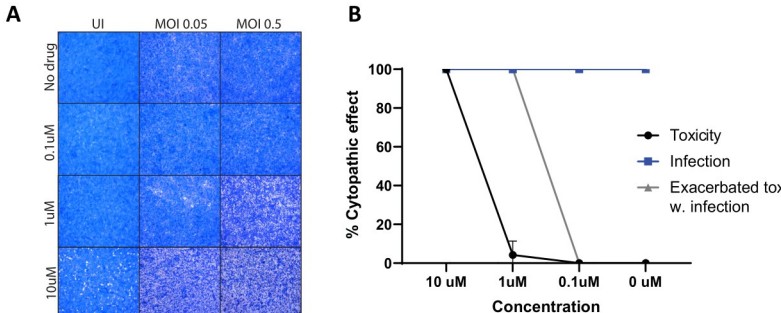

**Fig 1. ED50/LD50 for RGRN-305.** Toxicity of RGRN-305 and efficacy on inhibition of SARS-CoV2 infection were assessed in VeroE6-hTMPRSS2 cells in an ED50%/LD50% assay. The VeroE6-hTMPRSS2 cells were preincubated with RGRN-305 (0.1 μM, 1.0 μM and 10 μM) or vehicle 1 hour prior to infection with SARS-CoV2 B.1. (FR4286) isolate at a multiplicity of infection of 0.05 and 0.5. Cytopathic effect was assessed 72 hours post infection in cells stained with crystal violet (CV) solution. (**A**) Infection and toxicity were quantified by microscopy. Infection with the SARS-CoV2 B.1. strain caused cytopathological changes in the cells that were stained by CV solution as visible CV-dense spots/areas under the microscope. The SARS-CoV2 B.1. strain did not cause lytic replication or cell death in VeroE6-hTMPRSS2 under these conditions. Drug toxicity caused cell death resulting in the detachment of the VeroE6-hTMPRSS2 cells. Toxicity (cell death) was visualized as CV-negative (blank) areas under the microscope, corresponding to the detachment of dead cells. (**B**) % cytopathic effect caused by toxicity in uninfected wells (black line), % cytopathic effect caused by SARS-CoV2 infection (blue line), and % cytopathic toxicity scored in infected wells (Grey). % cytopathic effect (infection and toxicity) was calculated on biological triplicates of 8 wells. Abbreviations: CV, crystal violet UI, uninfected.

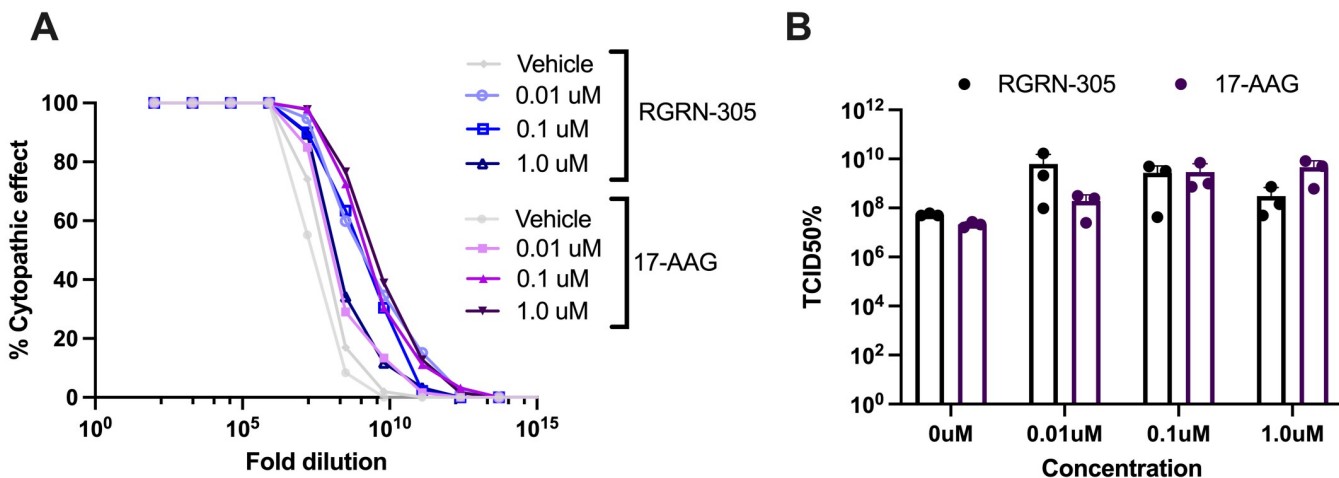

**Fig 2. TCID 50% on release of viral progeny.** The release of new viral progeny from SARS-CoV-2 infected VeroE6-hTMPRSS2 cells treated with RGRN-305 or 17-AAG. The VeroE6-hTMPRSS2 cells were pretreated with vehicle, RGRN-305 or 17-AAG (0.01 μM, 0.1 μM and 1.0 μM) 1 hour prior to infection with B.1. (FR4286) isolate at a multiplicity of infection of 0.05. After 48 hours, the supernatant was collected and the release of virus was quantified by a Tissue Culture Infectious Dose (TCID$_{50}$) assay and the Reed-Muench method. The release of SARS-CoV-2 was increased by RGRN-305 and 17-AAG treatment.

pretreated with RGRN-305, 17-AAG or vehicle for 1 hour prior to infection with SARS-CoV-2. Following 48 hours of infection, the supernatant was collected and the release of virus was quantified by a TCID50 assay, in which the cytopathic effects of a serial of dilutions of the virus were measured. The RGRN-305 and 17-AAG treatment increased the release of progeny SARS-CoV-2 virus particles to the cell supernatant in VeroE6-hTMPRSS2 cells (Fig 2).

## HSP90 inhibition by RGRN-305 dampened the cytokine activation of *TNF*, *IL1β* and *IL6* in A549 and U937 cell lines

To investigate the effects of HSP90 inhibition on inflammation induced by SARS-CoV-2, human macrophage-like cells U937 and human lung epithelial cells A549 were pretreated with RGRN-305 (5 μM) or water (drug-vehicle) for 8 hours prior to stimulation with SARS-CoV-2 S protein. Interestingly, RGRN-305 significantly inhibited the induction of *TNF*, *IL1β* and *IL6* in response to SARS-CoV-2 S protein, when measuring the mRNA expression by RT-qPCR (Fig 3).

To confirm the effects on protein level, the secreted levels of TNF, IL-1β and IL-6 were measured with ELISA. In this experiment, 17-AAG was used as a comparative reference. The secreted protein levels were consistent with the gene expression results (Fig 4). RGRN-305 and 17-AAG treatment strongly reduced the secretion of TNF, IL-1β and IL-6 in the U937 cell line, with a less pronounced effect in the A549 cell line. For the A549 cells, the secreted levels of IL-1β were undetectable and hence unavailable for evaluation. Taken together, HSP90 inhibition by RGRN-305 attenuated the cytokine activation elicited by SARS-CoV-2 S protein in U937 and A549 cell lines.

## HSP90 inhibition by RGRN-305 did not increase survival in a murine model of SARS-CoV-2

Transgenic mice expressing human ACE2 were inoculated with 1.5 x 10³ PFU of SARS-CoV-2 virus and treated with RGRN-305 (40 mg/kg body weight) or drug-vehicle once daily by oral gavage administration. The RGRN-305 treated mice exhibited reduced survival by the end of

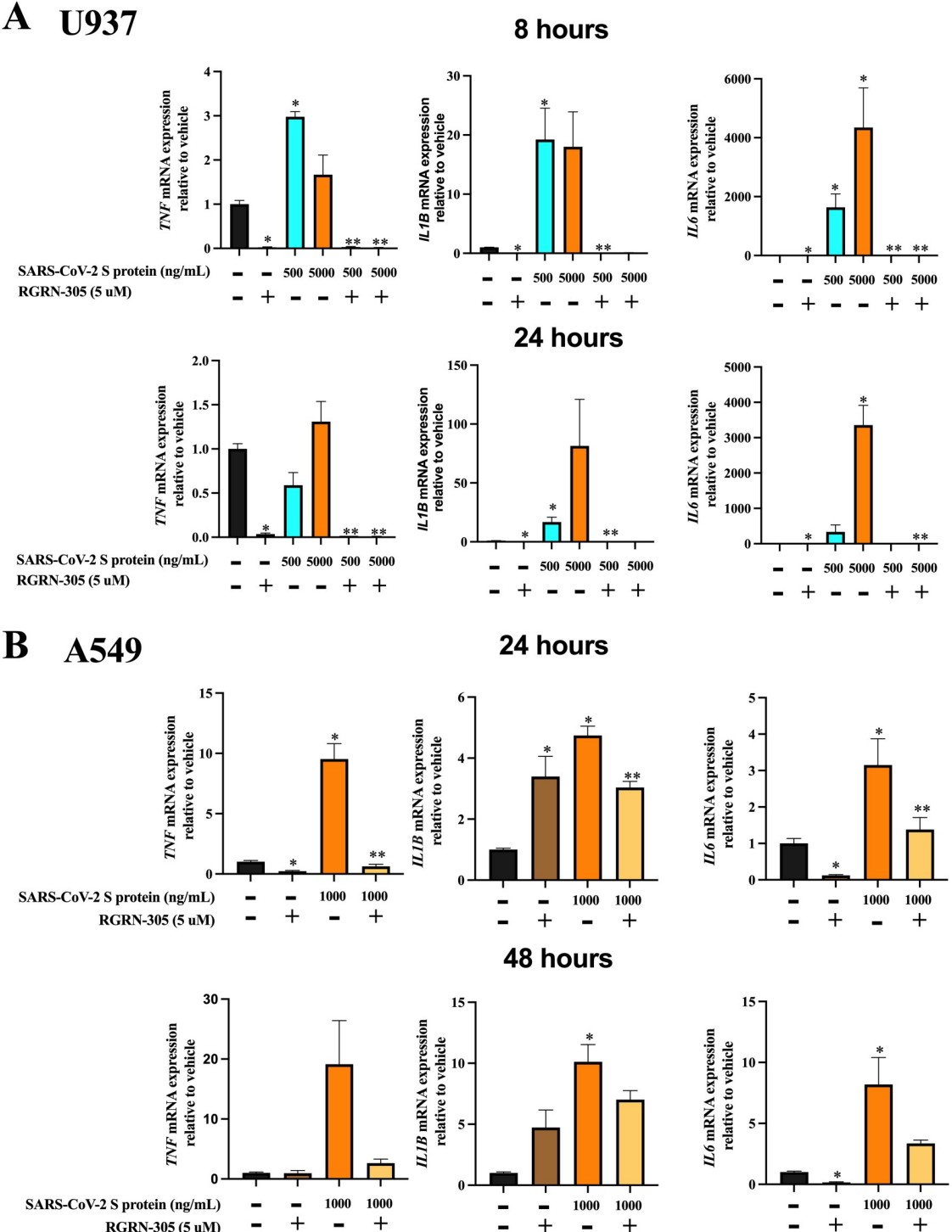

**Fig 3. HSP90 inhibition by RGRN-305 decreased the gene expression of proinflammatory cytokines induced by SARS-CoV-2 S protein.** RGRN-305 (HSP90 inhibitor) inhibited the mRNA expression of *TNF*, *IL1β* and *IL6* induced by SARS-CoV-2 S protein in human macrophage-like U937 and human lung epithelial A549 cells. **(A)** U937 macrophage-like cells were pretreated with RGRN-305 (5 μM) or water (drug-vehicle) for 8 hours prior to stimulation with SARS-CoV-2 S protein at a concentration of 500 ng/mL and 5000 ng/mL. After 8- and 24-hours post-stimulation, the mRNA expression of *TNF*, *IL1β* and *IL6* was measured by RT-qPCR (n = 4) **(B)** A549 cells were pretreated with RGRN-305 or water (drug-vehicle) for 8 hours before stimulation with SARS-CoV-2 S protein at a concentration of 1000 ng/mL for 24 and 48 hours. The expression of the indicated cytokines was measured by RT-qPCR (n = 5). Data represent mean ± SEM. * p < 0.05 compared to vehicle. ** p < 0.05 compared to corresponding stimulation without preincubation with RGRN-305.

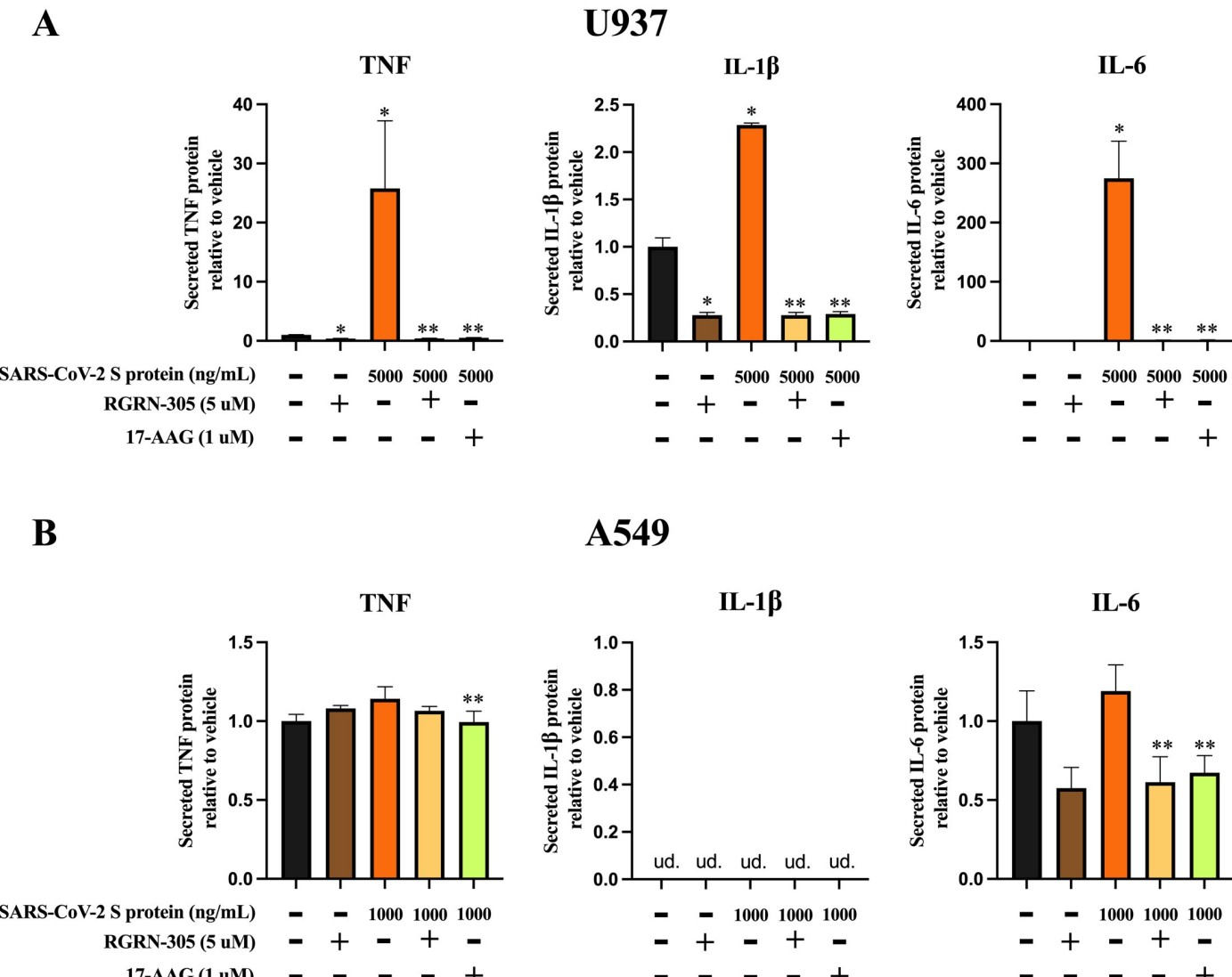

**Fig 4. HSP90 inhibition by RGRN-305 and 17-AAG decreased proinflammatory protein secretion.** RGRN-305 and 17-AAG reduced the secretion of TNF, IL-1β and IL-6 in U937 and A549 cells challenged with SARS-CoV-2 S protein. **(A)** U937 cells and **(B)** A549 cells were pretreated with drug-vehicle (water), RGRN-305 (5 μM) or 17-AAG (1 μM) for 8 hours followed by SARS-CoV-2 S protein challenge (U937, 5000 ng/ml; A549, 1000 ng/ml) for 24 hours. The supernatant was collected and the secreted protein expression of TNF, IL-1β and IL-6 was measured with ELISA. Data are shown as mean ± SEM. * $p < 0.05$ compared to vehicle. ** $p < 0.05$ compared to corresponding SARS-CoV-2 S protein stimulation without preincubation with RGRN-305 or 17-AAG.

the experiment (day 12) as 14% (1/7) survived compared to 63% (5/8) of those treated with -drug-vehicle (Fig 5A). Although, the differences in the Kaplan-Meier survival curves were not statistically significant (p = 0.0566), the data suggest that inhibition of HSP90 increases the mortality of mice infected with SARS-CoV-2. The weight of the mice was not significantly different between mice treated with RGRN-305 and drug-vehicle, however survivorship bias might have overestimated the weight of RGRN-305-treated mice (Fig 5B).

## Discussion

The inhibition of HSP90 has previously been shown to reduce the replication of SARS-CoV-2 and was proposed to be a promising treatment strategy for COVID-19 disease [7]. Conversely,

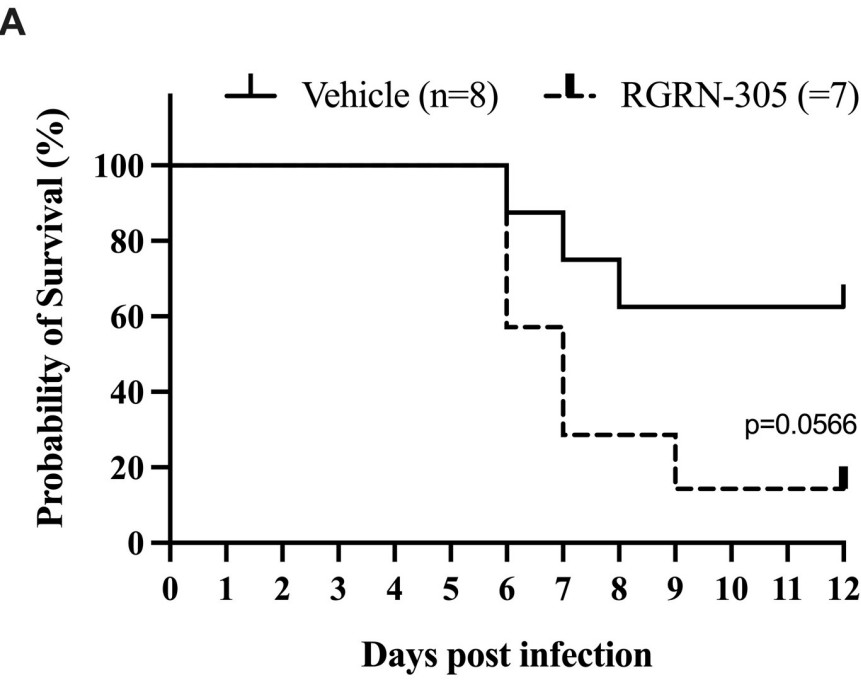

**Fig 5. Effect of SARS-CoV-2 infection on mortality and body weight of K18-hACE C57BL/6J mice treated with RGRN-305 or vehicle. (A-B)** Mice were inoculated with $1.5*10^3$ PFU of SARS-CoV-2 and treated once daily with 40 mg/kg body weight RGRN-305 (n = 7) or vehicle (n = 8) by oral gavage till the end of the experiment (day 12). **(A)** Kaplan-Meier survival curves of the mice treated with RGRN-305 (dashed line) and vehicle (black line) **(B)** The relative change of body weight compared to the weight before inoculation with SARS-CoV-2 of mice treated with RGRN-305 (black squares) compared to mice treated with vehicle (white squares). The dotted line corresponds to the 20% bodyweight reduction human endpoint criterion for euthanasia. Abbreviations: n.s., not significant. PFU, plaque-forming unit.

our data suggest that HSP90 inhibition by RGRN-305 and 17-AAG may exacerbate the cyto-pathic effect and increase the viral shedding in VeroE6-hTMPRSS2 cells infected with SARS-CoV-2 (Figs 1 and 2). In addition, HSP90 inhibition by RGRN-305 did not demonstrate a positive impact on survival and indeed there was an increased mortality in mice exposed to SARS-CoV-2. Nonetheless, RGRN-305 reduced the expression of *TNF*, *IL1B* and *IL6* in A549 and U937 cell lines, which may be useful against a cytokine storm. Systemic corticosteroids, such as dexamethasone, have beneficial effects in patients with severe COVID-19 disease requiring supplemental oxygen; however, they may have harmful effects (higher mortality, longer hospitalization and viral shedding) in patients with mild and moderate COVID-19 disease [16,17]. In agreement, the WHO guidelines state that corticosteroids should not be used, and may be harmful, in non-severe COVID-19 disease [18]. This suggests that immunosuppression may have a negative impact on early and mild-to-moderate COVID-19 disease. It is debatable how strong the elicited cytokine storm was in our mouse model [19–21]. Given the relatively low inoculation dose of virus used to infect our mice, the high survival (5/8) and the low body-weight reduction in our vehicle group, it is likely that our model did not elicit an extreme 'cytokine storm' associated with severe disease. Thus, the immunosuppressive effects of RGRN-305 and exacerbation of viral activity may be attributable to the lack of survival benefit by RGRN-305 in the SARS-CoV-2-infected mice. While the anti-inflammatory effects of RGRN-305 might be beneficial in mitigating the excessive immune activation in severe COVID-19 disease, further research is needed to investigate the immunomodulatory potential of RGRN-305 in combination with anti-viral drugs.

Li. *et al.* showed that Huh7 cells treated with 17-AAG potently suppressed the replication of both SARS-CoV and SARS-CoV-2 [12]. Similarly, 17-AAG caused efficient inhibition of SARS-CoV-2 replication (approximately 80% inhibition) while maintaining cell viability in Calu-3 cells [11]. Another HSP90 inhibitor, SNX-5422, inhibited the replication of SARS-CoV-2 in a dose-dependent manner by reducing viral RNA copies and infectious viral titer in the supernatant of infected Vero E6 and Calu-3 cells [14]. Furthermore, Wyler *et. al.* investigated the effect of three different HSP90 inhibitors (Onalespib, Ganetespib, 17-AAG) on SARS-CoV-2 replication in Calu-3 cells [10]. The viral yield in the supernatant and intracellular viral RNA was reduced by approximately 40%-70% depending on the HSP90 inhibitor [10]. These findings suggest that HSP90 proteins may facilitate SARS-CoV-2 replication. However, in our study, inhibition of HSP90 by RGRN-305 and 17-AAG resulted in increased viral replication. This discrepancy could stem from differences in experimental conditions, such as virus strain, multiplicity of infection, incubation times, cell lines and other variables. Alternatively, it's also possible that HSP90 plays a role in aiding the cell's stress response during viral infection, and disrupting this function may explain the heightened cytopathic effects and increased viral replication.

Nonetheless, HSP90 inhibitors may differ in their ability to inhibit SARS-CoV-2 replication due to differences in molecular structure, mode of inhibition, isoform selectivity, off-target effects and functions beyond the inhibition of the foldase activity. Hence, it may be of particular interest to investigate whether other HSP90 inhibitors that have been shown to effectively suppress SARS-CoV-2 replication *in vitro* may reduce viral replication, lung inflammation and mortality in mice infected with SARS-CoV-2. In a Syrian hamster model with SARS-CoV-2 GFP/ΔN, the HSP90 inhibitor 17-DMAG significantly reduced COVID-19 lung injury and gene expression of *TNF*, *IL1B* and *IL6*, but clinical deterioration was observed (weight loss exceeding 20% body weight and ruffled fur) [9]. Of note, this COVID-19 model did not manage to replicate a successful infection in hamsters but rather a single-cell infection, hence these findings may not be comparable to models with authentic SARS-CoV-2 infection [9]. However, these findings are consistent with ours, suggesting that while HSP90 inhibition may

reduce the SARS-CoV-2-induced inflammation, it may potentially lead to a worsened clinical outcome.

The findings of our study showed that inhibition of HSP90 by RGRN-305 dampened the inflammatory response induced by SARS-CoV-2 S protein in A549 and 937 cell lines by reducing the gene expression and secretion of specific proinflammatory cytokines such as TNF, IL-1β and IL-6. RGRN-305 strongly attenuated the secretion of these cytokines beyond normalcy (i.e., vehicle-treated cells) and to similar levels as 17-AAG-treated cells, suggesting that HSP90 inhibition provides a potent anti-inflammatory effect. In accordance, other studies have found anti-inflammatory effects of HSP90 inhibitors in Calu-3 cells and primary human airway epithelial cells infected with SARS-CoV-2 [10,14]. In line with this, HSP90 inhibition diminished pulmonary and systemic inflammation in a murine model of sepsis [22]. Additionally, RGRN-305 has shown to exhibit anti-inflammatory properties in other inflammatory models beyond virology, suggesting the role of HSP90 in inflammation involves multiple pathways [23–26].

Two patients in a clinical trial were infected with SARS-CoV-2 while receiving 250 mg RGRN-305 once daily, causing mild-to-moderate disease that resolved without any treatment [27]. The incidence of upper respiratory infections was similar between RGRN-305- and placebo-treated subjects. Furthermore, the mice were administered a dose of RGRN-305 that was approximately 13 times higher than the dose used in the human trial (40 mg/kg body weight corresponding to 3200 mg for an 80 kg person). These findings might indicate that 250 mg of RGRN-305 once daily presents an acceptable safety in patients with SARS-CoV-2 infection.

In summary, our data demonstrate that HSP90 inhibition by RGRN-305 did not exhibit antiviral activity against SARS-CoV-2 *in vitro* and no benefit to survival in mice infected with SARS-CoV-2. Nonetheless, the RGRN-305 treatment displayed a robust anti-inflammatory effect *in vitro* which may be useful in combination with anti-viral drugs like Remdesivir to treat severe COVID-19 disease with cytokine storms. Further animal studies are warranted to investigate whether HSP90 inhibitors may continue to be a potential treatment strategy for COVID-19 disease.

## Supporting information

**S1 Appendix. Datasets.**
(XLSX)

## Author Contributions

**Conceptualization:** Hakim Ben Abdallah, Manja Idorn, Line S. Reinert, Anne Bregnhøj, Søren Riis Paludan, Claus Johansen.

**Data curation:** Hakim Ben Abdallah, Giorgia Marino, Manja Idorn, Line S. Reinert.

**Formal analysis:** Hakim Ben Abdallah, Manja Idorn, Line S. Reinert.

**Funding acquisition:** Hakim Ben Abdallah, Claus Johansen.

**Investigation:** Hakim Ben Abdallah, Giorgia Marino, Manja Idorn, Line S. Reinert.

**Methodology:** Hakim Ben Abdallah, Giorgia Marino, Manja Idorn, Line S. Reinert, Anne Bregnhøj, Søren Riis Paludan, Claus Johansen.

**Project administration:** Anne Bregnhøj, Claus Johansen.

**Software:** Hakim Ben Abdallah, Manja Idorn.

**Supervision:** Anne Bregnhøj, Søren Riis Paludan, Claus Johansen.

**Writing – original draft:** Hakim Ben Abdallah, Giorgia Marino, Manja Idorn, Line S. Reinert.

**Writing – review & editing:** Hakim Ben Abdallah, Giorgia Marino, Manja Idorn, Line S. Reinert, Anne Bregnhøj, Søren Riis Paludan, Claus Johansen.

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
