## [Decision Letter · Decision Letter 0]

17 Jun 2024

PONE-D-24-13908Heat shock protein 90 inhibitor attenuates SARS-CoV-2 spike protein-induced inflammation * in vitro * but lacks effectiveness as COVID-19 treatment in mice***PLOS ONE*

Dear Dr. Ben Abdallah,

*Thank you for submitting your manuscript to PLOS ONE. After careful consideration, we feel that it has merit but does not fully meet PLOS ONE’s publication criteria as it currently stands. Therefore, we invite you to submit a revised version of the manuscript that addresses the points raised during the review process.*

*Please submit your revised manuscript by Aug 01 2024 11:59PM. If you will need more time than this to complete your revisions, please reply to this message or contact the journal office at plosone@plos.org. *

*Please include the following items when submitting your revised manuscript:*

*A rebuttal letter that responds to each point raised by the academic editor and reviewer(s). You should upload this letter as a separate file labeled 'Response to Reviewers'.*

*A marked-up copy of your manuscript that highlights changes made to the original version. You should upload this as a separate file labeled 'Revised Manuscript with Track Changes'.*

*An unmarked version of your revised paper without tracked changes. You should upload this as a separate file labeled 'Manuscript'.*

**

*We look forward to receiving your revised manuscript.*

*Kind regards,*

*Vinay Kumar, Ph.D.*

Academic Editor

*PLOS ONE*

*Journal Requirements:*

   "I have read the journal's policy and the authors of this manuscript have the following competing interests: 

H.B.A has received research grants from Aage Bangs Fund and participated in clinical trials sponsored by Galderma, Regranion and UCB. C.J. has served as a paid speaker for LEO Pharma, Eli Lilly and L’Oréal."

**

Reviewers' comments:

*Reviewer's Responses to Questions*

*

**Comments to the Author**
*

1. Is the manuscript technically sound, and do the data support the conclusions?

*The manuscript must describe a technically sound piece of scientific research with data that supports the conclusions. Experiments must have been conducted rigorously, with appropriate controls, replication, and sample sizes. The conclusions must be drawn appropriately based on the data presented. *

*Reviewer #1: Partly*

*Reviewer #2: No*

*2. Has the statistical analysis been performed appropriately and rigorously? *

*Reviewer #1: Yes*

*Reviewer #2: Yes*

*3. Have the authors made all data underlying the findings in their manuscript fully available?*

*The PLOS Data policy requires authors to make all data underlying the findings described in their manuscript fully available without restriction, with rare exception (please refer to the Data Availability Statement in the manuscript PDF file). The data should be provided as part of the manuscript or its supporting information, or deposited to a public repository. For example, in addition to summary statistics, the data points behind means, medians and variance measures should be available. If there are restrictions on publicly sharing data—e.g. participant privacy or use of data from a third party—those must be specified.*

*Reviewer #1: Yes*

*Reviewer #2: Yes*

*4. Is the manuscript presented in an intelligible fashion and written in standard English?*

*PLOS ONE does not copyedit accepted manuscripts, so the language in submitted articles must be clear, correct, and unambiguous. Any typographical or grammatical errors should be corrected at revision, so please note any specific errors here.*

*Reviewer #1: Yes*

*Reviewer #2: Yes*

*5. Review Comments to the Author*

*Please use the space provided to explain your answers to the questions above. You may also include additional comments for the author, including concerns about dual publication, research ethics, or publication ethics. (Please upload your review as an attachment if it exceeds 20,000 characters)*

*Reviewer #1: The authors investigated the role of RGRN-305, an HSP90 inhibitor, in VeroE6-hTMPRSS2 cells infected with severe acute respiratory syndrome coronavirus 2 (SARS-CoV-2) and in a murine model of SARS-CoV-2. They observed that inhibiting HSP90 exacerbated the cytopathic effect and did not reduce viral shedding in VeroE6-hTMPRSS2 cells. Furthermore, transgenic mice orally treated with an HSP90 inhibitor exhibited reduced survival compared to those treated with the drug vehicle. RGRN-305 also significantly dampened the inflammatory response induced by SARS-CoV-2 spike protein in human macrophage-like cells (U937) and human lung epithelial cells (A549). Overall, while RGRN-305 may be helpful in managing a cytokine storm, it has no beneficial impact on viral replication or survival in animals as a monotherapy. The manuscript is well-written and scientifically sound.*

Comments:

1. The authors have used "HSP90 inhibitor" as an alternative name for RGRN-305. Since there are many HSP90 inhibitors available with certain functional differences, the authors should use "RGRN-305" consistently, including in the title.

2. A novel finding of the manuscript is that RGRN-305 increased SARS-CoV-2 replication in VeroE6-hTMPRSS2 cells (Figure 2), unlike any other HSP90 inhibitor studied earlier, as mentioned in the discussion. However, the authors should use a well-known HSP90 inhibitor widely studied, like 17-allylamino-17-demethoxy-geldanamycin (17-AAG), as a control to demonstrate the difference in viral replication.

3. The authors also need to clarify that the effect of RGRN-305 is independent of its ability to inhibit HSP60, in comparison to 17-AAG as a positive control.

4. Another major point put forward by the authors is that RGRN-305 dampens the cytokine expression of TNFα, IL1β, and IL6 in A549 and U937 cell lines (Figure 3). Measuring mRNA expression by RT-qPCR to report the induction of TNFα, IL1β, and IL6 in response to SARS-CoV-2 S protein is too rudimentary, and standard ELISA assays should be used. Again, a positive control such as 17-AAG should be used for comparison.

5. The discussion section (Lines 317-337) should be revised to discuss why RGRN-305 might lead to increased SARS-CoV-2 replication independent of its HSP90 inhibition activity, compared to other well-known inhibitors.

*6. Throughout the manuscript, the authors should refrain from using broad statements like "HSP90 inhibitor" and specify the effects observed with "RGRN-305" in multiple instances. This includes changing the manuscript's title and rewriting the abstract.*

*Reviewer #2: In the present paper by Hakim Ben Abdallah et al, demonstrated that HSP90 inhibitor RGRN-305 exacerbates the SARS-CoV-2 infection in vitro and reduces the survival of mice infected with SARS-CoV-2, but exhibits strong anti-inflammatory properties. This suggested that the mentioned drug has a potential anti-inflammatory effects in COVID-19.*

I do have the following concerns and suggestions:

Major Points:

1. What is the rationale of using different concentrations of RGRN-305? Different concentrations were used in all different experiments. How these concentrations mimic physiological concentrations?

2. Currently there are numerous vaccines are available to combat the infection and the number of cases is reduced. This HSP90 inhibitor only showed anti-inflammatory effects and no effects on viral replication. If the authors are predicting the variants of sars-cov-2 may cause the spread of infection then how this inhibitor will be specifically combat with the infection and will be used as a therapeutic drug for COVID-19?

3. RGRN-305 is a specific inhibitor of HSP90? Does it show any effects against the isoforms of HSP90?

4. On one end authors are saying that this inhibitor has no effects on viral replication. On other end page 3; line 65 authors are saying ‘HSP90 inhibition 66 has been suggested to be a potential novel therapeutic strategy for COVID-19 treatment’. Justify the statement.

5. The quality of the images is poor. Submit the images with better DOI.

6. To strengthen the effects of HSP90 inhibitor on cytokine storm, authors have to show it with additional experiment like ELISA or Blotting.

7. Why the authors have used different concentrations of sars-cov-2 s protein in two cell lines? What is the rationale of using these specific concentrations?

8. Why there is a mortality in the drug-vehicle treated mice? Is there any other infection which triggers the inflammation and cause the death of the mice?

9. How effective is RGRN-305 compared to other HSP90 inhibitor available in the market?

Minor points:

1. Provide the catalogue numbers of the reagents and kits mentioned in the manuscript.

2. Page 3; line 51: Provide reference to the statement.

*3. Page 5; line 97: pr. To per*

*6. PLOS authors have the option to publish the peer review history of their article (what does this mean?). If published, this will include your full peer review and any attached files.*

**

*Reviewer #1: No*

*Reviewer #2: **Yes: **Satish K Raut*

**

*While revising your submission, please upload your figure files to the Preflight Analysis and Conversion Engine (PACE) digital diagnostic tool, https://pacev2.apexcovantage.com/. PACE helps ensure that figures meet PLOS requirements. To use PACE, you must first register as a user. Registration is free. Then, login and navigate to the UPLOAD tab, where you will find detailed instructions on how to use the tool. If you encounter any issues or have any questions when using PACE, please email PLOS at figures@plos.org. Please note that Supporting Information files do not need this step.*

---

## [Author Response · Author response to Decision Letter 0]

26 Aug 2024

Reviewer #1: The authors investigated the role of RGRN-305, an HSP90 inhibitor, in VeroE6-hTMPRSS2 cells infected with severe acute respiratory syndrome coronavirus 2 (SARS-CoV-2) and in a murine model of SARS-CoV-2. They observed that inhibiting HSP90 exacerbated the cytopathic effect and did not reduce viral shedding in VeroE6-hTMPRSS2 cells. Furthermore, transgenic mice orally treated with an HSP90 inhibitor exhibited reduced survival compared to those treated with the drug vehicle. RGRN-305 also significantly dampened the inflammatory response induced by SARS-CoV-2 spike protein in human macrophage-like cells (U937) and human lung epithelial cells (A549). Overall, while RGRN-305 may be helpful in managing a cytokine storm, it has no beneficial impact on viral replication or survival in animals as a monotherapy. The manuscript is well-written and scientifically sound.

Comments:

1. The authors have used "HSP90 inhibitor" as an alternative name for RGRN-305. Since there are many HSP90 inhibitors available with certain functional differences, the authors should use "RGRN-305" consistently, including in the title.

Authors response

Thank you for this valid point. We have specified that the HSP90 inhibition was achieved by RGRN-305 in the title and throughout the manuscript. 

2. A novel finding of the manuscript is that RGRN-305 increased SARS-CoV-2 replication in VeroE6-hTMPRSS2 cells (Figure 2), unlike any other HSP90 inhibitor studied earlier, as mentioned in the discussion. However, the authors should use a well-known HSP90 inhibitor widely studied, like 17-allylamino-17-demethoxy-geldanamycin (17-AAG), as a control to demonstrate the difference in viral replication.

Authors response

We agree with your good suggestion. We have performed new experiments with RGRN-305 and 17-AAG to examine the SARS-CoV-2 replication. Interestingly, we observed an increased viral replication by both RGRN-305 and 17-AAG in this model. We have revised the manuscript accordingly including Figure 2, methods, result and discussion sections. 

3. The authors also need to clarify that the effect of RGRN-305 is independent of its ability to inhibit HSP60, in comparison to 17-AAG as a positive control.

Authors response

We have assumed that the comment refers to potential functions independent on HSP90 inhibition. As suggested, we have added to the discussion section that HSP90 inhibitors may differ in their ability to inhibit SARS-CoV-2 replication due to differences in molecular structure, mode of inhibition, isoform selectivity, off-target effects and functions beyond the inhibition of the foldase activity (L405-L407).

4. Another major point put forward by the authors is that RGRN-305 dampens the cytokine expression of TNFα, IL1β, and IL6 in A549 and U937 cell lines (Figure 3). Measuring mRNA expression by RT-qPCR to report the induction of TNFα, IL1β, and IL6 in response to SARS-CoV-2 S protein is too rudimentary, and standard ELISA assays should be used. Again, a positive control such as 17-AAG should be used for comparison. 

Authors response

Thank you for this suggestion. As recommended, we have repeated the experiments with 17-AAG and measured the secretion of TNF, IL-1β and IL-6 using ELISA assays. Interestingly, both RGRN-305 and 17-AAG strongly suppressed the secretion of these cytokines in U937 cells, and to a lesser degree in A549 cells (Figure 4). The secreted levels of IL-1β in A549 cells were undetectable and not possible for evaluation. The manuscript has been revised accordingly.

5. The discussion section (Lines 317-337) should be revised to discuss why RGRN-305 might lead to increased SARS-CoV-2 replication independent of its HSP90 inhibition activity, compared to other well-known inhibitors.

Authors response

Thank you for this feedback. We showed that both RGRN-305 and 17-AAG increased the SARS-CoV-2 replication. As suggested, we have revised this discussion section to include a discussion of the discrepancies in the literature (L384-L418). 

6. Throughout the manuscript, the authors should refrain from using broad statements like "HSP90 inhibitor" and specify the effects observed with "RGRN-305" in multiple instances. This includes changing the manuscript's title and rewriting the abstract.

Authors response

The suggestions have been implemented throughout the manuscript. 

Reviewer #2: In the present paper by Hakim Ben Abdallah et al, demonstrated that HSP90 inhibitor RGRN-305 exacerbates the SARS-CoV-2 infection in vitro and reduces the survival of mice infected with SARS-CoV-2, but exhibits strong anti-inflammatory properties. This suggested that the mentioned drug has a potential anti-inflammatory effects in COVID-19.

I do have the following concerns and suggestions:

Major Points:

1. What is the rationale of using different concentrations of RGRN-305? Different concentrations were used in all different experiments. How these concentrations mimic physiological concentrations?

Authors response

Thank you for these relevant questions. 

In the first SARS-CoV-2 experiment (Figure 1), we used three concentrations of RGRN-305 (0.1 µM, 1 µM, 10 µM) to determine the toxicity in uninfected and infected VeroE6-hTMPRSS2 cells. RGRN-305 was toxic to the uninfected cells at 10 μM, whereas 0.1 μM and 1 μM were tolerated. Thus, we continued to use concentrations ranging from 0.1 to 1 µM in the subsequent SARS-CoV 2 experiments with VeroE6-hTMPRSS2 cells (Figure 2). 

In the experiments with A549 and U937 challenged with SARS-CoV 2 S protein, we used 5 µM of RGRN-305 because we routinely use this concentration to assess anti-inflammatory effects and this concentration was well-tolerated by the A549 and U937 cell lines.

The used concentrations are in range with physiological concentrations. In a published study (doi: 10.1093/annonc/mdv031), cancer patients received doses ranging from 50 mg to 1600 mg of RGRN-305 leading to plasma concentrations ranging between 0.05 to 0.7 µM. However, it’s worth noting that RGRN-305 accumulates in tissue, leading to higher tissue concentrations than plasma concentrations. In humanized mouse models (xenotransplanted), tissue concentrations (1-5 µM) have shown to be 5-10 fold higher than plasma concentrations (doi: 10.1158/1078-0432.CCR-09-0152, doi: 10.2340/00015555-1838).

2. Currently there are numerous vaccines are available to combat the infection and the number of cases is reduced. This HSP90 inhibitor only showed anti-inflammatory effects and no effects on viral replication. If the authors are predicting the variants of sars-cov-2 may cause the spread of infection then how this inhibitor will be specifically combat with the infection and will be used as a therapeutic drug for COVID-19?

Authors response

Thank you for your comment. We fully agree with your assessment. We have concluded that RGRN-305 provided no anti-viral effects and led to a worsened clinical outcome in mice, thus it may have no utility as a monotherapy for COVID-19. We believe this is an important finding because HSP90 inhibition has previously been evaluated in other in vitro studies to be a potential drug target for COVID-19. 

Nonetheless, it is worth noting that RGRN-305 (and 17-AAG) provided strong anti-inflammatory effects in our study, which may be useful in combination with anti-viral drugs like Remdisvir to treat COVID-19 disease with cytokine storms. However, further studies are needed to investigate this. 

3. RGRN-305 is a specific inhibitor of HSP90? Does it show any effects against the isoforms of HSP90?

Authors response

Thank you for highlighting this point. In response, we have included the IC50 values for RGRN-305 against the isoforms in the methods sections (L86-L87).

4. On one end authors are saying that this inhibitor has no effects on viral replication. On other end page 3; line 65 authors are saying ‘HSP90 inhibition 66 has been suggested to be a potential novel therapeutic strategy for COVID-19 treatment’. Justify the statement.

Authors response

We apologize for this misunderstanding. We have specified that recent studies from the literature suggest that HSP90 inhibition may be a novel therapeutic strategy for COVID-19 treatment (L72-L73).

5. The quality of the images is poor. Submit the images with better DOI.

Authors response

We have submitted revised images with improved quality. 

6. To strengthen the effects of HSP90 inhibitor on cytokine storm, authors have to show it with additional experiment like ELISA or Blotting.

Authors response

Thank you for your good suggestion. We have repeated the A549 and U937 experiments and included the HSP90 inhibitor 17-AAG as a comparative reference (Figure 4). Please see our response to Comment 4 by Reviewer 1 for more information. 

7. Why the authors have used different concentrations of sars-cov-2 s protein in two cell lines? What is the rationale of using these specific concentrations?

Authors response

In preliminary experiments (n = 2), four concentrations of the SARS-CoV-2 S protein (100, 500, 1000, and 5000 ng/mL) were tested. The optimal concentrations were selected based on their ability to induce inflammatory gene expression and minimize cell toxicity. This has been added to the methods sections (L185-L188). 

8. Why there is a mortality in the drug-vehicle treated mice? Is there any other infection which triggers the inflammation and cause the death of the mice?

Authors response

We selected an inoculation dose at which we anticipated that the drug-vehicle-treated mice would succumb to the SARS-CoV-2 infection. Given that the primary clinical endpoint was mortality, a lethal inoculation dose was required. 

9. How effective is RGRN-305 compared to other HSP90 inhibitor available in the market?

Authors response

This is a relevant discussion. There are no head-to-head clinical studies to answer this. The IC50 values presented for RGRN-305 indicate that it’s a strong HSP90 inhibitor. We have included 17-AAG as a comparative reference in our repeated experiments with infected VeroE6-hTMPRSS2 cells to evaluate the SARS-CoV 2 replication (Figure 2) and with U937/A549 cells to evaluate the anti-inflammatory effects. RGRN-305 and 17-AAG demonstrated similar effects. We have revised our discussion section to include a discussion of other HSP90 inhibitors and potential differences (L384-L437).

Minor points:

1. Provide the catalogue numbers of the reagents and kits mentioned in the manuscript.

Authors response

The catalogue numbers for relevant reagents have been provided in the manuscript such as the HSP90 inhibtors (RGRN-305 and 17-AAG), cell lines, SARS-CoV-2 Spike protein, TPA, PCR Primers/Probes, ELISA kits etc. 

2. Page 3; line 51: Provide reference to the statement.

Authors response

Reference 3 is cited: 

‘The clinical presentation of SARS-CoV-2 infection ranges from asymptomatic and mild upper respiratory symptoms to pneumonia and multiorgan failure, which can be fatal [3].’

3. Page 5; line 97: pr. To per

Authors response

This has been added to the manuscript.

---

## [Decision Letter · Decision Letter 1]

9 Sep 2024

The heat shock protein 90 inhibitor RGRN-305 attenuates SARS-CoV-2 spike protein-induced inflammation in vitro but lacks effectiveness as COVID-19 treatment in mice

PONE-D-24-13908R1

Dear Dr. Ben Abdallah,

We’re pleased to inform you that your manuscript has been judged scientifically suitable for publication and will be formally accepted for publication once it meets all outstanding technical requirements.

Kind regards,

Vinay Kumar, Ph.D.

Academic Editor

PLOS ONE

Additional Editor Comments (optional):

Reviewers' comments:

Reviewer's Responses to Questions

**Comments to the Author**

1. If the authors have adequately addressed your comments raised in a previous round of review and you feel that this manuscript is now acceptable for publication, you may indicate that here to bypass the “Comments to the Author” section, enter your conflict of interest statement in the “Confidential to Editor” section, and submit your "Accept" recommendation.

Reviewer #1: All comments have been addressed

Reviewer #2: All comments have been addressed

2. Is the manuscript technically sound, and do the data support the conclusions?

Reviewer #1: Yes

Reviewer #2: Yes

3. Has the statistical analysis been performed appropriately and rigorously? 

Reviewer #1: Yes

Reviewer #2: Yes

4. Have the authors made all data underlying the findings in their manuscript fully available?

Reviewer #1: Yes

Reviewer #2: Yes

5. Is the manuscript presented in an intelligible fashion and written in standard English?

Reviewer #1: Yes

Reviewer #2: Yes

6. Review Comments to the Author

Reviewer #1: I am satisfied with response to my comments from the authors and the manuscript had improvised considerably.

Reviewer #2: (No Response)

7. PLOS authors have the option to publish the peer review history of their article (what does this mean?). If published, this will include your full peer review and any attached files.

Reviewer #1: **Yes: **Sannigrahi MK

Reviewer #2: No

---

## [Editor Report · Acceptance letter]

16 Sep 2024

PONE-D-24-13908R1 

PLOS ONE

Dear Dr. Ben Abdallah, 

I'm pleased to inform you that your manuscript has been deemed suitable for publication in PLOS ONE. Congratulations! Your manuscript is now being handed over to our production team.

Kind regards, 

on behalf of

Dr. Vinay Kumar 

Academic Editor

PLOS ONE